# What’s in a Smile? Initial Analyses of Dynamic Changes in Facial Shape and Appearance [note 1]

**DOI:** 10.3390/jimaging5010002

**Published:** 2018-12-21

**Authors:** Damian J. J. Farnell, Jennifer Galloway, Alexei I. Zhurov, Stephen Richmond, David Marshall, Paul L. Rosin, Khtam Al-Meyah, Pertti Pirttiniemi, Raija Lähdesmäki

**Affiliations:** 1School of Dentistry, Cardiff University, Heath Park, Cardiff CF14 4XY, UK; 2School of Computer Science and Informatics, Cardiff University, Cardiff CF24 3AA, UK; 3Research Unit of Oral Health Sciences, Faculty of Medicine, University of Oulu, FI-90014 Oulu, Finland; 4Medical Research Center Oulu (MRC Oulu), Oulu University Hospital, FI-90014 Oulu, Finland

**Keywords:** multilevel principal components analysis, shape and image texture, facial expression

## Abstract

Single-level principal component analysis (PCA) and multi-level PCA (mPCA) methods are applied here to a set of (2D frontal) facial images from a group of 80 Finnish subjects (34 male; 46 female) with two different facial expressions (smiling and neutral) per subject. Inspection of eigenvalues gives insight into the importance of different factors affecting shapes, including: biological sex, facial expression (neutral versus smiling), and all other variations. Biological sex and facial expression are shown to be reflected in those components at appropriate levels of the mPCA model. Dynamic 3D shape data for all phases of a smile made up a second dataset sampled from 60 adult British subjects (31 male; 29 female). Modes of variation reflected the act of smiling at the correct level of the mPCA model. Seven phases of the dynamic smiles are identified: rest pre-smile, onset 1 (acceleration), onset 2 (deceleration), apex, offset 1 (acceleration), offset 2 (deceleration), and rest post-smile. A clear cycle is observed in standardized scores at an appropriate level for mPCA and in single-level PCA. mPCA can be used to study static shapes and images, as well as dynamic changes in shape. It gave us much insight into the question “what’s in a smile?”.

## 1. Introduction

Human faces are central to our identity and they are important in expressing emotion. The act of smiling is important in this context and the exploration of facial changes and dynamics during the act of smiling [1,2] is an ongoing topic of investigation in fields of research in orthodontics and prosthodontics, both of which aim to improve the function and appearance of dentition. Aesthetics (e.g., of smiles [3]) are therefore an important aspect of these fields. Much research into the “science of a smile” also focuses on the effects of aging and biological sex on the shape and appearance [4,5] and also the dynamics [6,7,8] of smiling. Recent investigations have been greatly enhanced by the use of three-dimensional (3D) imaging techniques [9,10,11,12,13] that allow both static and dynamic imaging of the face. Clinically, this research has led to improved understanding of orthognathic surgery [9], malocclusion [10], associations between facial morphology and cardiometabolic risk [11], lip shape during speech [12], facial asymmetry [13], and sleep apnea [14] (to name but a few examples). Clearly also, facial simulation is of much interest for human–computer interfaces (see, e.g., [15,16,17]). The role of genetic factors on facial shape has also been the subject of much recent attention [18,19,20,21] and many factors (sex, age, and genetic factors) across a set of subjects can affect the shape and dynamics of facial and/or mouth shape.

In this article, we wish to explore the question “what’s in a smile?” by using multilevel principal components analysis (mPCA) to adjust for covariates such as natural mouth shape and/or sex. Indeed, the mPCA approach has previously been shown [22,23,24,25,26] to provide a simple and straightforward method of modeling shape. The mPCA approach is potentially also of much use in active shape models (ASMs) [27,28,29,30,31] and active appearance models (AAMs) [32,33,34,35,36,37] (see also [38]). We remark that one such previous application of mPCA to ASMs related to the segmentation of the human spine [22]. The authors stated that their results showed that “such a modelization offers more flexibility and allows deformations that classical statistical models can simply not generate”. They also noted that “the idea is to decompose the data into a within-individual and a between-individual component”. Hence, we remark again that mPCA provides one method of adjusting for external and/or confounding factors or covariates that can strongly affect shapes (or images). Other recent applications of mPCA [23,24,25] allowed us to determine the relative importance of biological sex and ethnicity on facial shape by examination of eigenvalues. Modes of variation made sense because changes in shape were seen to correspond to biological sex and ethnicity at the correct levels of the model and no ‘mixing’ of these effects was observed. Finally, principal component ‘scores’ also showed strong clustering, which were again at the correct levels of the mPCA model.

Another method that allows us to investigate the effects of covariates on facial shape to be modeled is called bootstrapped response-based imputation modeling (BRIM) [20,21]. The effects of covariates such as sex and genomic ancestry on facial shape were summarized in [20] as response-based imputed predictor (RIP) variables and the independent effects of particular alleles on facial features were uncovered. Indeed, the importance of modeling the effects of covariates in images is also becoming increasingly recognized, e.g., such as in variational auto-encoders (see, e.g., [39,40,41]) in which the effects of covariates are modeled as latent variables sandwiched between encoding (convolution) and decoding (deconvolution) layers. However, the topic of variational auto-encoders lies beyond the scope of this article. Linear discriminant functions have also been used previously (see, e.g., [42,43]) to explore groupings in the subject population for image data.

The work presented here is also an expansion of [26] that extended the mPCA approach from shape data also to include image data, where a set of (frontal) facial images from a group of 80 Finnish subjects (34 male; 46 female) each for two different facial expressions (smiling and neutral) were considered. A three-level model illustrated by Figure 1 was constructed that contains biological sex, facial expression, and ‘between-subject variation’ at different levels of the model and we use this model again here for this dataset. However, we also compare results of mPCA to those results of single-level PCA, which was not carried out in [26]. Furthermore, the dynamic aspects of a smile are considered here in a new (time series) dataset of 3D mouth shape captured during all phases of a smile, which was also not considered in [26]. We present here firstly the subject characteristics and details of image capture and preprocessing. We then consider the mathematical detail of the mPCA method. We then present our results for both datasets. Finally, we offer a discussion of these results in the concluding section.

## 2. Materials and Methods

### 2.1. Image Capture, Preprocessing, and Subject Characteristics

Dataset 1 consisted of photographic images of the 80 adult Finnish subjects (34 female; 46 male) who were captured with two expressions (neutral and smiling). Patients were from the Northern Finland Birth Cohort NFBC-66 (http://www.oulu.fi/nfbc/) and all patients were 46 years old when the images were collected. The Ethical Committee of the Northern Ostrobothnia Hospital District, Oulu University Hospital, Oulu, Finland has approved this study. Twenty-one reliable facial landmarks, illustrated by Figure 2, were positioned manually for each image and these points are used here in the analysis of shape. Preprocessing of the shapes included centering, alignment, and adjustment of overall scale only. Preprocessing of image texture [26] also included the definition of a region of interest (ROI) around edges of the face (7339 pixels) and the overall illumination of the grayscale images was standardized.

Dataset 2 consisted of 3D video shape data during all phases of a smile, where 13 points placed (and tracked) along the outer boundary of mouth. Subjects were 60 adult staff and students at Cardiff University (31 male and 29 female). The number of frames in the video was between approximately 100 and 250 frames during all phases of a smile for each subject. In these initial calculations, preprocessing consisted of centering the 3D shapes only. The (normalized) smile amplitude was defined by using the following equation.
(1)$$\mathrm{Smile}\;\mathrm{Amplitude}=\frac{\mathrm{Distance}\;\mathrm{between}\;\mathrm{the}\;\mathrm{left}\;\mathrm{and}\;\mathrm{right}\;\mathrm{chelion}\;\mathrm{at}\;\mathrm{time}\;t}{2\times \mathrm{Distance}\;\mathrm{between}\;\mathrm{the}\;\mathrm{left}\;\mathrm{and}\;\mathrm{right}\;\mathrm{chelion}\;\mathrm{at}\;\mathrm{time}\;t=0}$$

Phases of the smile (rest pre-smile, onset acceleration, onset deceleration, apex, offset acceleration, offset deceleration, and rest post-smile) were identified manually by inspection of time series of smile amplitudes for each subject individually. This is illustrated schematically in Figure 3. Acceleration and deceleration phases are divided by points of inflexion in the amplitude. Inspection also of the (smoothed) first and second derivatives of the smile amplitude with respect to time allowed accurate estimation of the boundaries between these phases.

### 2.2. Multilevel Principal Components Analysis (mPCA)

ASMs [26,27,28,29,30] are statistical models of shape only and AAMs [32,33,34,35,36,37] are statistical models of both shape and appearance. The term ‘image texture’ is taken to refer to the pattern of intensities or colors across an image (or image patch) as in AAMs and we adopt this usage here. For ASMs, features may be segmented from an image by firstly forming the shape model over some ‘training’ set of shapes. Single-level principal component analysis (PCA) may be used to define the distributions of points or intensities (and/or color) at given pixel positions, respectively. For single-level PCA of shape, the mean shape vector (averaged over all *N* subjects) is given by $\bar{z}$ and a covariance matrix, *C*, is then found by evaluating
(2)$$C_{k_{1},k_{2}}=\frac{1}{N-1}\sum_{i=1}^{N}\left(z_{ik_{1}}-{\bar{z}}_{ik_{1}}\right)(z_{ik_{2}}-{\bar{z}}_{ik_{2}}),$$
where $k_{1}$ and $k_{2}$ indicate elements of this covariance matrix. We find the eigenvalues $\lambda_{l}$ and (orthonormal) eigenvectors $u_{l}$ of this matrix. All eigenvalues are non-negative, real numbers because covariance matrices are symmetric and positive semi-definite. For PCA, one ranks all of the eigenvalues $\lambda_{l}$ into descending order and we retain the first $l_{1}$ components in the model. Any new shape is then modeled by
(3)$$z^{model}=\bar{z}+\sum_{l=1}^{l_{1}}a_{l}u_{l},$$

The coefficients, $a_{l}$, for a fit of the model to a new shape vector, z, are found readily by using a scalar product with respect to the set of orthonormal eigenvectors $u_{l}$, i.e.,
(4)$$a_{l}=u_{l}\cdot (z-\bar{z}).$$

A similar process is carried out for image texture or appearance [32,33,34,35,36,37]. For ASMs, features of interest in a new image can then be segmented by obtaining a trial shape from an image (e.g., defined along strong edges), which is then projected into the shape model in order to find an improved estimate that is consistent with the model via some constraint on the coefficients, $a_{l}$. This process is iterated until convergence and so the final segmentation never ‘strays too far’ from a plausible solution with respect to the underlying shape model. In this article, we are concerned only with the modeling of shape and appearance and we do not carry out the ‘active’ image searches. However, the methods presented here could be extended to such image searches via ASMs or AAMs, although this is not the primary focus of this article.

Multilevel PCA (mPCA) allows us to isolate the effects of various influences on shape or image texture at different levels of the model. For the case of the act of “smiling”, this allows us to adjust for each subjects’ individual shape or appearance (and biological sex also for dataset 1) in order can get a clearer picture of these general changes due to a primary factor (here, facial expression due to smiling). This is illustrated schematically in Figure 1 for dataset 1. Multiple levels are used in mPCA to model the data and covariance matrices are formed at each level. For the model for dataset 1 illustrated by Figure 1, the covariance matrix at level 3 is formed with respect to the two expressions (neutral smiling) for each subject and then these covariance matrices are averaged over all 80 subjects. By contrast, the covariance matrix at level 2 is formed with respect to shapes or image texture averaged over the two expressions for each of the 80 subjects. Covariance matrices are formed for males and females separately and then they are averaged over two sexes to find the final covariance matrix at level 2 of this model. Finally, the covariance matrix at level 2 is formed with respect to shapes or image texture sex only, i.e., averaged over all subjects and expressions for the two sex groups separately. The number of shapes or images equals 2 at this level of the model and so the rank of this matrix is 1. Clearly, any restriction on the rank of the covariance matrix is a limitation of the mPCA model, although other multilevel methods (such as multilevel Bayesian approaches) ought not to be as strongly constrained as mPCA. An exploration of these topics will form the contents of future research. A three-level mPCA model was also used in this initial exploration for dataset 2: level 1, “between subject” due to natural face shape not attributed to smiling; level 2, “between smile phases” variation due to differences between seven different phases of a smile; level 3, “within smile phases” variation due to residual differences within the different phases of a smile. Hence, we explicitly model subjective variation across participants at specific levels of the mPCA model for both datasets.

mPCA then uses PCA with respect to these covariance matrices at each of the three levels separately. The *l*-th eigenvalue at level 1 is denoted by $\lambda_{l}^{1}$ with associated eigenvector $u_{l}^{1}$, whereas the *l*-th eigenvalue at level 2 is denoted by $\lambda_{l}^{2}$ with associated eigenvector is denoted by $u_{l}^{2}$, and so on. We rank all of the eigenvalues into descending order at each level of the model separately, and then we retain the first $l_{1}$, $l_{2}$, and $l_{3}$ eigenvectors of largest magnitude for all three levels, respectively. Any new shape is modeled by
(5)$$z^{model}=\bar{z}+\sum_{l=1}^{l_{1}}a_{l}^{1}u_{l}^{1}+\sum_{l=1}^{l_{2}}a_{l}^{2}u_{l}^{2}+\sum_{l=1}^{l_{3}}a_{l}^{3}u_{l}^{3},$$
where $\bar{z}$ is now the ‘grand mean’. The coefficients $\left\{a_{l}^{1}\right\}$, $\left\{a_{l}^{2}\right\}$, and $\left\{a_{l}^{3}\right\}$ (also referred to here as ‘component scores’) are determined for any new shape, z, by using a global optimization procedure in MATLAB R2017 with respect to the cost function
(6)$$\Delta =\sum_{k=1}^{}(z_{k}-z_{k}^{model}{)}^{2}=\sum_{k=1}^{}(z_{k}-{\bar{z}}_{k}-\sum_{l=1}^{l_{1}}a_{l}^{1}u_{lk}^{1}-\sum_{l=1}^{l_{2}}a_{l}^{2}u_{lk}^{2}-\sum_{l=1}^{l_{3}}a_{l}^{3}u_{lk}^{3}{)}^{2}.$$
note again that $z_{k}$ is the *k*-th element of the shape vector z, and $u_{lk}^{1}$ indicates the *k*-th element of the *l*-th eigenvector at level 1, etc. Another method of obtaining a solution is to iterate the following equations directly,
(7)$$a_{l}^{\alpha }\leftarrow a_{l}^{\alpha }-\kappa \frac{\partial \Delta }{\partial a_{l}^{\alpha }},$$
(where α = 1, 2, 3, here and for all values *l* appropriately) for our three-level models from some ‘starting point’ (often taken to be the average shape, i.e., all coefficients are zero) until convergence. An appropriate choice of *κ* is found to be *κ* = 0.01. It is straightforward to see that
(8)$$\frac{\partial \Delta }{\partial a_{l}^{\alpha }}=-2\sum_{k}u_{lk}^{\alpha }(z_{k}-{\bar{z}}_{k}-\sum_{m=1}^{l_{1}}a_{m}^{1}u_{mk}^{1}-\sum_{m=1}^{l_{2}}a_{m}^{2}u_{mk}^{2}-\sum_{m=1}^{l_{3}}a_{m}^{3}u_{mk}^{3}).$$

Note that this approach finds identical solutions to that provided by MATLAB. Finally, standardized coefficients may be found by dividing the $\left\{a_{l}\right\}$ coefficients by the square root of the corresponding eigenvalue $\lambda_{l}$ for single-level PCA and by dividing the $\left\{a_{l}^{1}\right\}$ coefficients by the square root of the corresponding eigenvalue $\lambda_{l}^{1}$ at level 1 for mPCA (and similarly for the other levels). These standardized coefficients or ‘scores’ are useful in exploring clustering that may occur by biological sex, facial expression, and so on.

## 3. Results

Eigenvalues for both shape and also image texture via mPCA are shown in Figure 4 for dataset 1. The results for mPCA demonstrated a single non-zero eigenvalue for the level 1 (biological sex). A single large eigenvalue for the level 3 (facial expression) for mPCA occurs for shape and also for image texture, although many non-zero (albeit of much smaller magnitude) eigenvalues occur for image texture. Level 2 (between-subject variation) via mPCA tends to have the largest number of non-zero eigenvalues for both shape and image texture. The first two eigenvalues are (relatively) large at level 2, mPCA for image texture. mPCA results suggest that biological sex seems be the least important for this group of subjects for both shape and texture, although caution needs to be exercised as the rank of the matrix is 1 at this level for both shape and texture. Results for the eigenvalues from single-level PCA are of comparable magnitude to those results of mPCA, as one would expect, and they follow a very similar pattern. Inspection of these results for the eigenvalues tell us broadly that facial expression and natural facial shape (not dependent on sex or expression) are strong influences on facial shapes in the dataset. Biological sex was found to be a weaker effect comparatively, especially for shape, although again caution needs to be exercised in interpreting eigenvalues at this level for mPCA.

Modes of variation of shape for dataset 1 are presented in Figure 5. The first mode at level 3 (facial expression) for mPCA and mode 1 in single-level PCA both capture changes in facial expression (i.e., neutral to smiling and vice versa). Changes in mouth shape in Figure 5 can be seen that relate clearly to the act of smiling in both graphs. For example, obvious effects such as widening of the mouth, corners of the mouth raised slightly, exposure of teeth can be clearly seen. However, subtle effects such as narrowing of the eyes [44] and a slight widening at bottom of nose during smile are also seen clearly, especially for mPCA. Eyes become further apart (relatively) and the mouth becomes wider for the first mode at level 1 (biological sex) for mPCA in Figure 4. All shapes have been scaled so that the average point-to-centroid distance is equal to 1 and so this result makes sense because men have generally thinner faces than women on average [45,46]. This first mode via mPCA at level 3 probably corresponds to mode 3 or mode 2 (or a combination of both) in single-level PCA. However, modes 2 and 3 from single-level PCA are harder to interpret, and one can never preclude the possibility of mixing of different influences (e.g., sex, expression, etc.) in modes in single-level PCA. By contrast, mPCA should focus more clearly on individual influences because they are modeled at different levels of the mPCA model. The first mode at level 2 (between-subject variation) for mPCA in Figure 5 (middle row) corresponds to changes in the relative thinness/width of the face (presumably) that can occur irrespective of sex.

Modes of variation for image texture for dataset 1 are presented in Figure 6. The first modes at each level are relatively straightforward to understand for mPCA. We see that mode 1 for level 1 (biological sex) mPCA does indeed correspond to changes in appearance due to biological sex (e.g., females tend to have more prominent eyes and cheeks [45,46]), as required. Mode 1 for level 2 (between-subject) mPCA corresponds to residual changes due to left/right position (possibly) and also illumination, although this mode is slightly harder to interpret. Mode 1 for level 3 (facial expression) mPCA corresponds to changes due to the act of smiling (i.e., mean − SD = not smiling, mean = half smile, and mean + SD = full smile). We see clear evidence of a smile that exposes the teeth in this mode. Furthermore, subtle effects are seen for mPCA at this level such as increased prominence of the cheeks, increased nose width, and narrowing of the eyes [44]. The first three modes for single-level PCA are also relatively straightforward to interpret, although arguably less so than for the first mode at each level from mPCA. For example, mode 1 possibly corresponds to residual changes in illumination and/or also to slight changes to the nose and prominence of the cheeks, which might be associated with biological sex [45,46]. Modes 2 and 3 correspond clearly to changes due to the act of smiling.

Results for the standardized component ‘scores’ for mPCA for shape are shown in Figure 7. Results for component 1 for single-level PCA demonstrate differences due to facial expression clearly because the centroids are strongly separated between smiling and neutral expressions. By contrast, component 2 for single-level PCA does not seem to reflect changes due to either facial expression or biological sex very strongly. Finally, component 3 for single-level PCA reflects differences due to biological sex (albeit mildly), as there is some separation in the centroids between males and females. The centroids in Figure 7 at level 1 (biological sex) for mPCA are strongly separated by biological sex, although not by facial expression. Hence, strong clustering by biological sex (alone) is observed for shape at level 1 (biological sex) for mPCA. The centroids in Figure 7 at level 3 (facial expression) for mPCA are strongly separated by facial expression (neutral, smiling), although not by biological sex. Strong clustering by facial expression (alone) is therefore observed at level 3 (facial expression) for mPCA, also as required. Strong clustering by facial expression or biological sex is not seen at level 2 (between-subject variation) mPCA (not shown here), i.e., all centroids by biological sex and facial expression are congruent on the origin.

Results for the standardized component ‘scores’ for mPCA for image texture are shown in Figure 8. Component 1 for single-level PCA reflects differences due to biological sex and components 2 and 3 reflect changes due to facial expression. Again, level 1 for mPCA clearly reflects differences by biological sex and level 3 reflects differences by expression (neutral or smiling). Again, strong clustering by facial expression or biological sex is not seen at level 2 (between-subject variation) for mPCA also for image texture (not shown here), i.e., all centroids by biological sex and facial expression are congruent on the origin.

Eigenvalues via single-level PCA and mPCA for dataset 2 (‘dynamic’ 3D shape data) are shown in Figure 9. A single large eigenvalue for the level 2 (facial expression/smiling) occurs for mPCA, although the following eigenvalues are relatively larger than for the analysis of shapes for dataset 1. All phases of a smile are captured here in the 3D video shape data in dataset 2 and this result is to be expected. Variation at level 1 (between-subject variation) for mPCA tends to have larger eigenvalues compared to those for levels 2 and 3 (facial expression/smiling). Level 3 eigenvalues via mPCA are found to be minimal, thus indicating that residual variations within each smile phase are small. Finally, results of single-level PCA are of similar magnitude and follow a similar pattern to those eigenvalues from mPCA.

Modes of variation of shape for dataset 2 are presented in Figure 10. Results for the first mode at level 1 (between-subject variation) via mPCA in the coronal plane correspond to changes between upturned and downturned lip shape, which is consistent with changes due to subjects’ natural lip shape. Results for the first mode at level 1, mPCA in the transverse plane appear to indicate changes in the prominence of the upper and lower lips. By contrast, results for the first mode of variation at level 2 (i.e., between smile phases level) for mPCA correspond to increased mouth size and a strong drawing back (and slight upturn) of the corners of the mouth, which is consistent with the act of smiling. Mode 1 from single-level PCA is broadly similar to mode 1 at level 1 (between-subject variation) via mPCA, whereas mode 2 from single-level PCA is broadly similar to mode 1 at level 2 (between smile phases) via mPCA. Results for the modes of variation via single-level PCA for dataset 2 are therefore not presented here.

Standardized component scores from both single-level PCA and mPCA at level 2 (variation between smile phases) with respect to shape for dataset 2 are shown in Figure 11. Very little difference between centroids divided by smile phase is seen at levels 1 or 3 for mPCA (not shown here). The centroids of component scores at level 2 via mPCA are clearly separated in Figure 11 with respect to the seven phases of a smile (i.e., rest pre-smile, onset 1 (acceleration), onset 2 (deceleration), apex, offset 1 (acceleration), offset 2 (deceleration), and rest post-smile). Indeed, we see clear evidence of a cycle in these centroids in Figure 11 over all of these smile phases for both single-level PCA and at level 2 for mPCA. These results are strong evidence that seven phases of a smile do indeed exist.

## 4. Discussion

We have shown in this article that mPCA provides a viable method of accounting for groupings in our population subject set and/or for adjusting for potential confounding covariates. For example, natural face or lip shape was represented at one level of the mPCA model and shapes changes due to the act of smiling at another level (or levels) of the model. By capturing these different sources of variation we represented at different levels of the model, we are able to isolate those changes in expression due to smiling that are consistent over the entire populate much more effectively than single-level PCA. All results were found to agree with results of single-level PCA, although mPCA results were (arguably) easier to interpret than those results of single-level PCA.

For the first dataset considered here that contained two ‘expressions’ per subject (neutral or smiling), both obvious effects (widening of the mouth, corners of mouth raised slightly, exposure of teeth, and increased prominence of cheeks), and subtle effects (narrowing of the eyes and a slight widening at bottom of nose during smile) were detected in major modes of variation for the facial expression level of the mPCA model. Inspection of eigenvalues suggested that facial expression and ‘between-subject’ effects were strong influences on shape and image texture, although biological sex was a weaker effect especially for shape. Indeed, another study [24] has noted that sexual dimorphism was weakest for a Finnish population in comparison to other ethnicities (i.e., English, Welsh, and Croatian). Furthermore, the first major mode for shape showed clearly that males have longer/thinner faces on average than women [45,46] at an appropriate level of the mPCA model. Changes in image texture also clearly corresponded to biological sex, again at an appropriate level of the mPCA model. Model fits gave standardized scores for each principal component/mode of variation that show strong clustering for both shape and texture by biological sex and facial expression also at appropriate levels of the model. mPCA correctly decomposes sources of variation due to biological sex and facial expression (etc.). These results are an excellent initial test of the usefulness of mPCA in terms of modeling either shape or image texture.

For the second dataset that contained 3D time-series shape data, results of major modes of variation via mPCA were seen to correspond to the act of smiling. Inspection of eigenvalues again showed that both ‘natural lip shape’ and facial expression are strong sources of shape variation, as one would expect. Previous studies of 3D facial shape changes have posited that there are three phases to a smile [8,15,16,17], namely, onset, apex, and offset. However, if one includes rests pre and post smiling, standardized component scores from both mPCA (at the appropriate level of the model) and single-level PCA demonstrated clear evidence of a cycle containing seven phases of a smile: rest pre-smile, onset 1 (acceleration), onset 2 (deceleration), apex, offset 1 (acceleration), offset 2 (deceleration), and rest post-smile. This is strong evidence that seven phases of a smile do indeed exist and it is another excellent test of the mPCA method, now also for dynamic 3D shape data.

Future research will focus on modeling the effects of ethnicity, gender, age, genetic information, or diseases (e.g., effects perhaps previously hidden in the ‘final 5% of variation’) on facial shape or appearance. The present study has not considered the effects of “outliers” in the shape or image data. Clearly, the effects of outliers (either as isolated points, subjects or indeed even entire groups of subjects) might strongly affect mean averages used to estimate centroids of groups and also covariance matrices at the various levels of the model. The simplest method of addressing this problem is to use robust centroid and covariance matrix estimation [47,48,49] and then to carry out PCA as normal at each level. Note that robust covariance matrix estimation is included in MATLAB (2017a) and so this may be implemented easily, although a sample size of at least twice the length of the feature vector *z* is required. Furthermore, the mPCA method uses averages of covariance matrices (e.g., over all subjects in the population or over specific subgroups) and robust averaging of these matrices might also be beneficial. Clearly also, we can use other forms of robust PCA [50,51,52] and *M*-estimators [53,54] might also to deal with the problem of outliers. Finally, future research will attempt to extend existing single-level probabilistic methods of modeling shape and/or appearance (e.g., mixtures models [30,55] and extensions of Bayesian methods used in ASMs or AAMs [56,57]) to multilevel formulations and to active learning [58]. The use of schematics such as Figure 1 will hopefully prove just as useful in visualizing these models as they have for mPCA.

## 5. Conclusions

We have demonstrated in this article that mPCA can be used to study dynamic changes of facial shape and appearance. Our analyses yielded great insight into the question “what’s in a smile?”. PCA-based methods have the advantage that one can obtain an idea of the importance of modes of variation by the inspection of eigenvalues. This also appears to be the case for mPCA, although more caution must be exercised as the rank of covariance matrices might be constrained at some levels of the model. Although eigenvectors are orthogonal to each other within a given level, it should be noted that they do not need to be orthogonal between levels for mPCA. Potentially, this could lead to difficulty in carrying out model fits when obtaining component ‘scores’, especially for small numbers of mark-up points, although this did not seem to be a problem here. However, it might be that certain covariates affect facial or mouth shape in ways that are, in fact, inherently non-orthogonal. mPCA method provides a way of addressing this issue, whereas we would strongly expect single-level PCA to mix effects between different covariates in the principal components in such cases.

## Figures and Tables

**Figure 1 jimaging-05-00002-f001:**
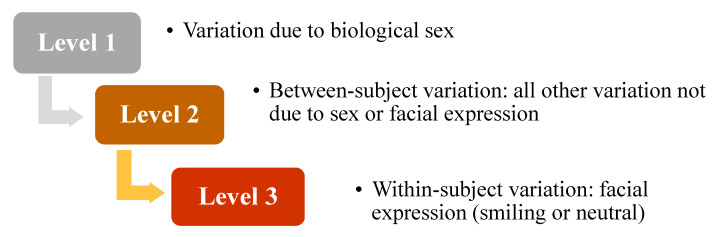
Flowchart illustrating the multilevel model of facial shape for dataset 1.

**Figure 2 jimaging-05-00002-f002:**
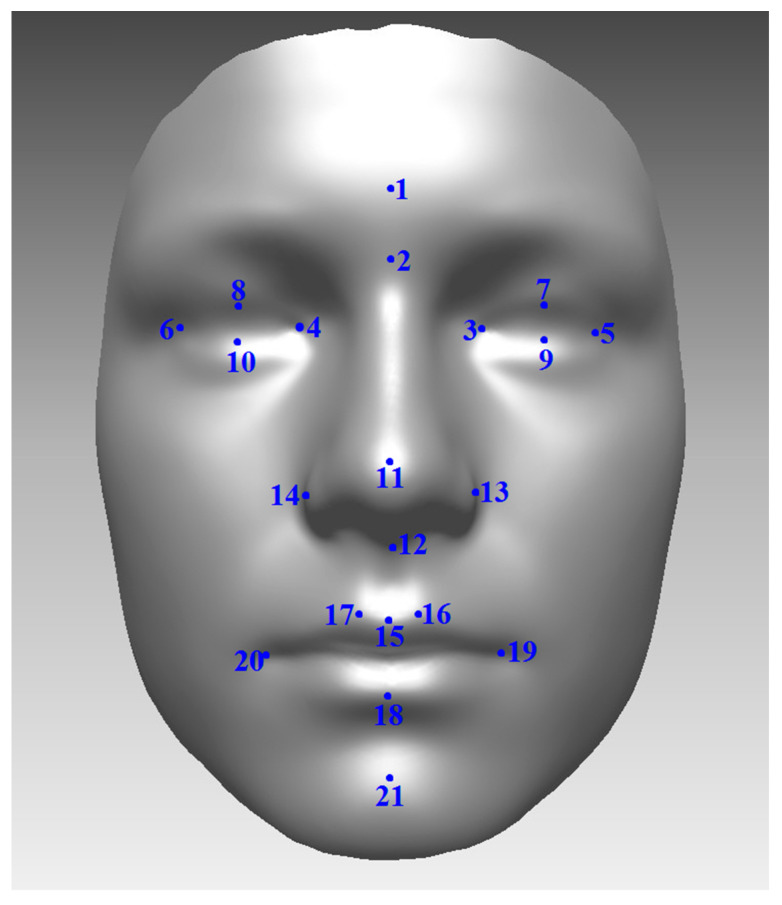
Illustration of the 21 landmark points for dataset 1 ((1) Glabella (g); (2) Nasion (n); (3) Endocanthion left (enl); (4) Endocanthion right (enr); (5) Exocanthion left (exl); (6) Exocanthion right (exr); (7) Palpebrale superius left (psl); (8) Palpebrale superius right (psr); (9) Palpebrale inferius left (pil); (10) Palpebrale in-ferius right (pir); (11) Pronasale (prn); (12) Subnasale (sn); (13) Alare left (all); (14) Alare right (alr); (15) Labiale superius (ls); (16) Crista philtri left (cphl); (17) Crista philtri right (cphr); (18) Labiale inferius (li); (19) Cheilion left (chl); (20) Cheilion right (chr); (21) Pogonion (pg)).

**Figure 3 jimaging-05-00002-f003:**
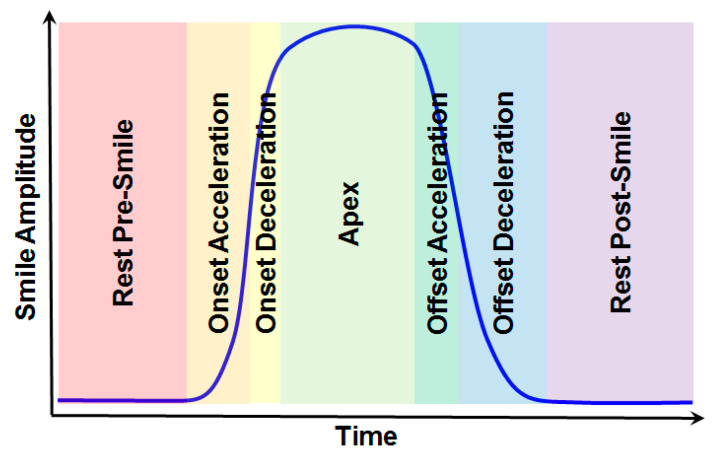
Schematic illustration of a time series of smile amplitudes from Equation (1) for 3D shape data in dataset 2. Including rest phases, seven phases can be identified manually: rest pre-smile, onset acceleration, onset deceleration, apex, offset acceleration, offset deceleration, and rest post-smile.

**Figure 4 jimaging-05-00002-f004:**
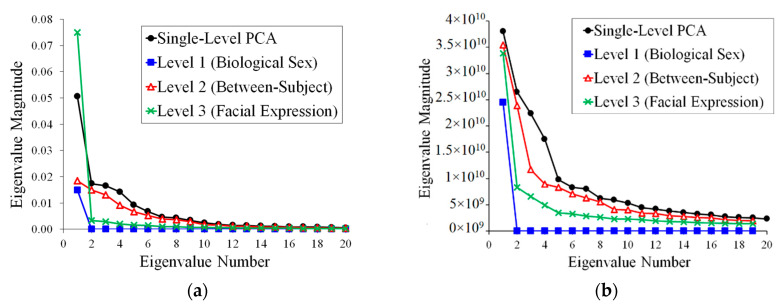
Eigenvalues for dataset 1 from single-level PCA and from mPCA level 1 (biological sex), level 2 (between-subject variation), and level 3 (within-subject variation: facial expression). (**a**) Shape data; (**b**) Image texture data (All shapes have been scaled so that the average point-to-centroid distance equals 1.).

**Figure 5 jimaging-05-00002-f005:**
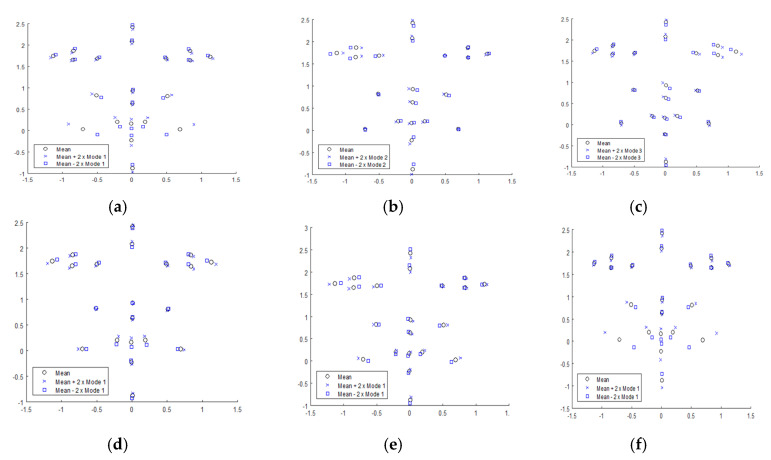
Modes of variation for shape for dataset 1 for the first three modes from single-level PCA in the upper set of images: (**a**) = mode 1; (**b**) = mode 2; (**c**) = mode 3. The first modes from levels 1 to 3 mPCA in the bottom set of images: (**d**) = mode 1, level 1 (biological sex); (**e**) = mode 1, level 2 (between subjects); (**f**) = mode 1, level 3 (facial expression).

**Figure 6 jimaging-05-00002-f006:**
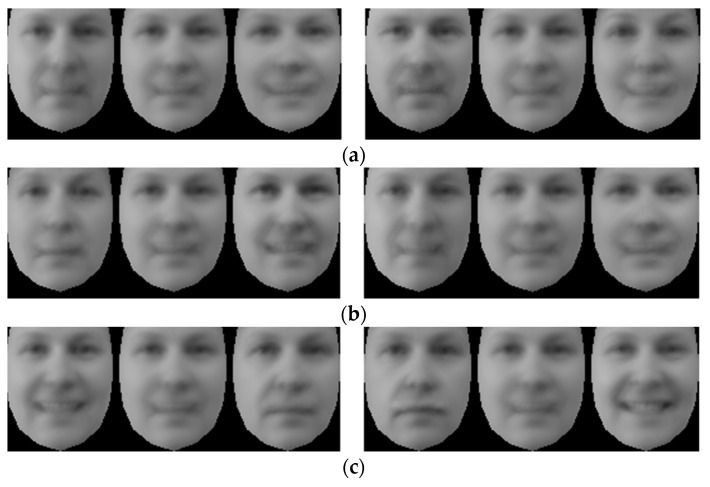
Modes of variation for image texture for dataset 1 for the first three modes ((**a**) = mode 1; (**b**) = mode 2; (**c**) = mode 3) from single-level PCA in the left-hand set of images, and the first modes from levels 1 to 3 ((**a**) = level 1; (**b**) = level 2; (**c**) = level 3) from mPCA in the right-hand set of images. Note that for each set of three images: left image = mean − SD; middle image = mean; right image = mean + SD.

**Figure 7 jimaging-05-00002-f007:**
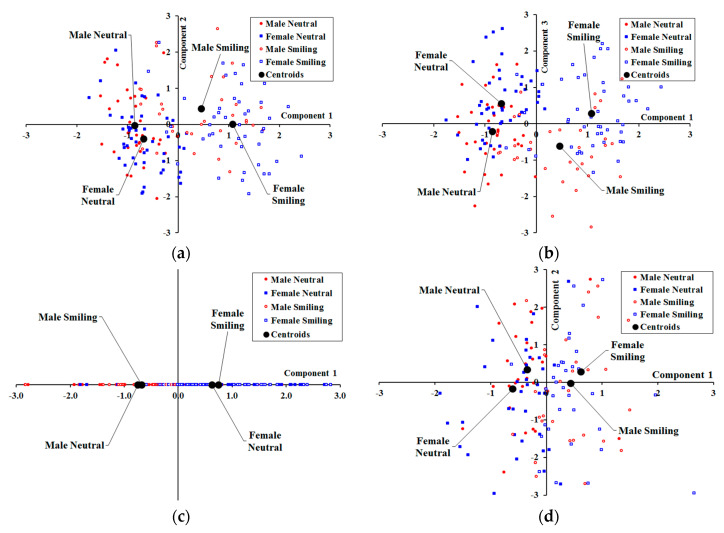
Standardized component scores with respect to shape for dataset 1: (**a**) Components 1 and 2 for single-level PCA; (**b**) Components 1 and 3 for single-level PCA; (**c**) Component 1 for level 1 (biological sex) for mPCA; (**d**) Components 1 and 2 for level 3 (facial expression) for mPCA.

**Figure 8 jimaging-05-00002-f008:**
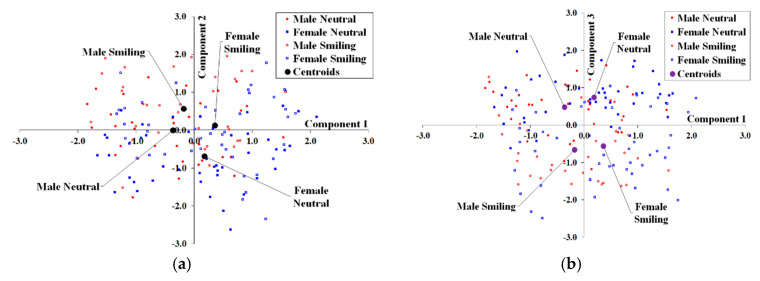
Standardized component scores with respect to image texture for dataset 1: (**a**) Components 1 and 2 for single-level PCA; (**b**) Components 1 and 3 for single-level PCA; (**c**) Component 1 for level 1 (biological sex) for mPCA; (**d**) Components 1 and 2 for level 3 (facial expression) for mPCA.

**Figure 9 jimaging-05-00002-f009:**
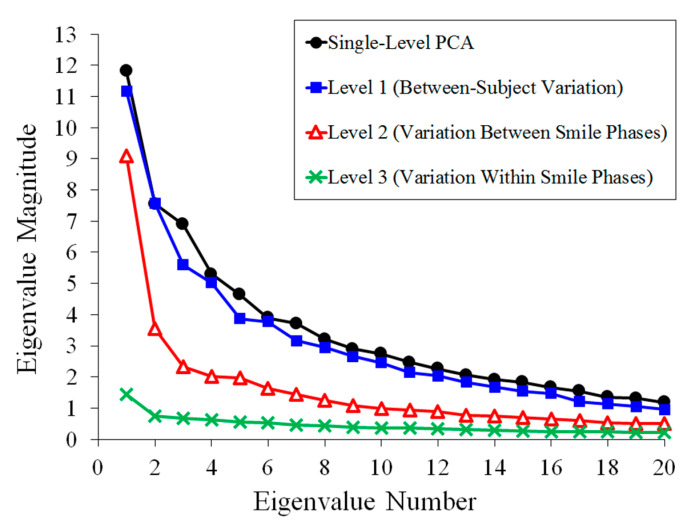
Eigenvalues for dataset 2 (shape data only) from single-level PCA and from mPCA level 1 (between-subject variation), level 2 (variation between smile phases), and level 3 (variation within smile phases).

**Figure 10 jimaging-05-00002-f010:**
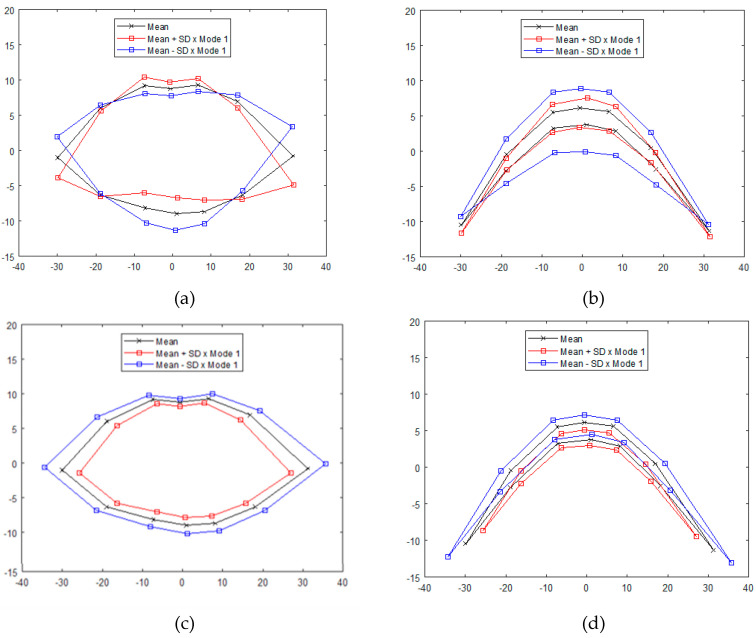
Modes of variation for dataset 2 from mPCA: (**a**) level 1 (between-subject variation), mode 1, coronal plane; (**b**); level 1 (between-subject variation), mode 1, transverse plane; (**c**) level 2 (variation between smile phases), mode 1, coronal plane; (**d**) level 2 (variation between smile phases), mode 1, transverse plane.

**Figure 11 jimaging-05-00002-f011:**
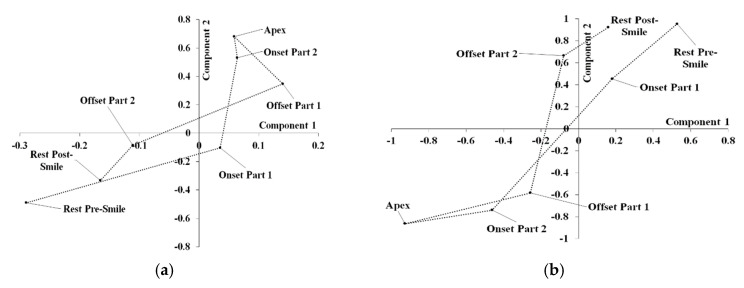
Centroids over smile phases for standardized component scores with respect to shape for dataset 2: (**a**) Components 1 and 2 for single-level PCA; (**b**) Components 1 and 2 at level 2 (variation between smile phases) for mPCA.

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
