# Peer review of "What’s in a Smile? Initial Analyses of Dynamic Changes in Facial Shape and Appearance†"

_2313-433X, 2018, doi:10.3390/jimaging5010002_

Round 1
Reviewer 1 Report
# good work and well written.
# Mathematical expressions have to be rewritten. In a few case, there are wrong.
# Experimental results are convincing.
# References are OK, and I suggest the following:
Sema Candemir, Eugene Borovikov, K. C. Santosh, Sameer K. Antani, George R. Thoma: RSILC: Rotation- and Scale-Invariant, Line-based Color-aware descriptor. Image Vision Comput. 42: 1-12 (2015)
# In case of active learning, where data has to consider real-time, authors would like to consider the following work to be discussed (future work):
Mohamed-Rafik Bouguelia, Slawomir Nowaczyk, K. C. Santosh, Antanas Verikas:
Agreeing to disagree: active learning with noisy labels without crowdsourcing. Int. J. Machine Learning & Cybernetics 9(8): 1307-1319 (2018)
Author Response
Response to referee in attached file.

Reviewer 2 Report
The paper is well done. The authors may want to split the conclusion into discussion and conclusion or rename the section as discussion and conclusion.
The authors should add some statistical analysis in the result section to compare the results across participants.
Author Response
Response to referee in attached file.
